# A label-efficient two-sample test

Weizhi Li[1]          Gautam Dasarathy[2]          Karthikeyan Natesan Ramamurthy[3]          Visar Berisha[2]

[1]School of Computing and Augmented Intelligence, Arizona State University, Tempe, Arizona, USA
[2]School of Electrical, Computer and Energy Engineering, Arizona State University, Tempe, Arizona, USA
[3]IBM Thomas J Watson Research Center, Yorktown Heights, NY, USA

## Abstract

Two-sample tests evaluate whether two samples are realizations of the same distribution (the null hypothesis) or two different distributions (the alternative hypothesis). We consider a new setting for this problem where sample features are easily measured whereas sample labels are unknown and costly to obtain. Accordingly, we devise a three-stage framework in service of performing an effective two-sample test with only a small number of sample label queries: first, a classifier is trained with samples uniformly labeled to model the posterior probabilities of the labels; second, a novel query scheme dubbed *bimodal query* is used to query labels of samples from both classes, and last, the classical Friedman-Rafsky (FR) two-sample test is performed on the queried samples. Theoretical analysis and extensive experiments performed on several datasets demonstrate that the proposed test controls the Type I error and has decreased Type II error relative to uniform querying and certainty-based querying. Source code for our algorithms and experimental results is available at https://github.com/wayne0908/Label-Efficient-Two-Sample.

## 1 INTRODUCTION

Two-sample hypothesis testing evaluates whether two samples (or sets of data points) are generated from the same distribution (null hypothesis) or different distributions (alternative hypothesis). A conventional two-sample test is formulated as follows [Johnson and Kuby, 2011]: (a) the statistician obtains two sets of data points $\mathcal{X} = \{x_1, , \ldots, x_{n_0}\}$ and $\mathcal{Y} = \{y_1, \ldots, y_{n_1}\}$; (b) she computes a test statistic $\mathcal{T}(\mathcal{X}, \mathcal{Y})$; (c) she then computes the $p$-value of the observed

test statistic under the null hypothesis (both $\mathcal{X}$ and $\mathcal{Y}$ come from the same distribution). A low $p$-value implies that, under the null hypothesis, observing a value for the statistic at least as extreme as the one observed is unlikely to happen, and the null hypothesis may be rejected.

To motivate our novel two-sample testing problem, we think of the observed data as being a set of measurements $S = \mathcal{X} \bigcup \mathcal{Y} = \{s_1, \ldots, s_n\}$ and a set of corresponding group labels $\mathcal{Z} = \{z_1, \ldots, z_n\}$, where $z_i = 0$ if $s_i \in \mathcal{X}$ and 1 otherwise. We think of the $s_i$'s as features and the set of $z_i$'s as the corresponding labels. Accordingly, our observation model is $n$ i.i.d draws from the joint distribution $p_{SZ}(s, z)$. The two sample testing problem under this formulation is equivalent to testing if $p_{S|Z}(\cdot \mid 0) = p_{S|Z}(\cdot \mid 1)$ (i.e., $S$ and $Z$ are independent).

In traditional two-sample testing (see e.g., Friedman and Rafsky [1979], Chen and Friedman [2017], Hotelling [1992], Friedman [2004], Clémençon et al. [2009], Lhéritier and Cazals [2018], Hajnal [1961]), the underlying assumption is that both the features and their corresponding labels are simultaneously available. In this paper, we extend two-sample hypothesis testing to a new and important setting where the measurements (or features) $s_1, \ldots s_n$ are readily accessible, but their groups (or labels) $z_1, \ldots z_n$ are unknown and difficult/costly to obtain. A good representative example is the validation of digital biomarkers in Alzheimer's disease relative to imaging markers. Say we want to determine whether a series of digital biomarkers (e.g. gait, speech, typing speed measured using a patient's smartphone) is related to amyloid buildup in the brain (measured from neuroimaging, and an indication of increased risk of Alzheimer's disease). In this scenario, we can obtain the digital biomarkers on a large scale by distributing the tests via the internet. However, actually determining if a particular patient is amyloid positive (higher risk of Alzheimer's disease) or negative (lower risk) involves expensive neurological imaging, and it is of considerable interest to reduce this cost. Notice that this scenario is in stark contrast to traditional formulations of two sample testing, where the class label (amyloid positivity) is assumed

*Accepted for the 38th Conference on Uncertainty in Artificial Intelligence* (UAI 2022).

to be readily available. This paper addresses this problem formulation by constructing a label-efficient two-sample test.

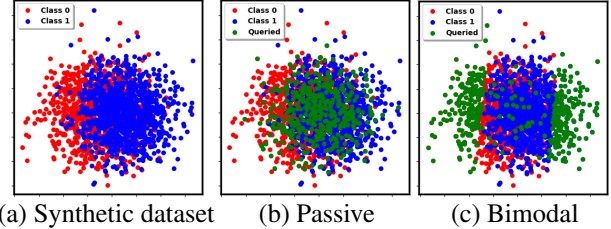

(a) Synthetic dataset     (b) Passive     (c) Bimodal

Figure 1: A synthetic dataset with two classes shown in **blue** and **red**, and queried datapoints shown in **green** returned by the passive query and the proposed bimodal query.

**Contributions** We propose a **three-stage** framework for label efficient two-sample hypothesis testing: in the **first stage**, we "model" the class probability (posterior probability) of a sample by training a classifier with a small set of uniformly sampled data; in the **second stage**, we propose a new query scheme dubbed *bimodal query* that queries the labels of samples with the highest posterior probabilities from both groups, and in the **third stage**, the classical Friedman and Rafsky (FR) two-sample test [Friedman and Rafsky, 1979] is performed on the queried samples to accept or reject the null-hypothesis. The intuition behind our framework is that the classifier trained on the uniformly sampled datapoints will identify the regions with most distributional difference between $p_{S|Z}(\cdot \mid 0)$ and $p_{S|Z}(\cdot \mid 1)$; these points are then labeled by an oracle. As a result, under the alternate being true, this procedure solves a different, much simpler version of the problem, thereby reducing the number of labeled samples required to reject the null. This is facilitated by the bimodal query scheme shown in Fig. 1. As is clear from the figure, when bimodal query (Fig. 1(c)) is used to label the samples, the points with maximum separation between the distributions are selected whereas the passive query (uniform sampling) maintains the original separation.

The query scheme is theoretically motivated by identifying an optimal marginal distribution $p_{q^*}(s)$ such that, under the alternative hypothesis, the test has increased power. That is, we derive the $p_{q^*}(s)$ that minimizes the asymptotic FR testing statistic. For samples that are i.i.d generated from $p_{q^*}(s)$, we further show that the convergence rate of a variant of the FR test statistic is independent of feature dimension $d$. Our query scheme approximates sampling from this distribution and we demonstrate that our framework can control the Type I error at a desired level when a permutation test is used. We empirically demonstrate increased power when our test is used on synthetic data, the MNIST dataset [LeCun, 1998], and a dataset from the Alzheimer's Disease Neuroimaging Initiative (ADNI) database [Jack Jr et al., 2008].

**Related literature** The problem setting considered in this paper is distinct from the previous work. While [Naghsh-var et al., 2013, Chernoff, 1959] propose active hypothesis testing, they actively select actions/experiments and generate both sample measurements (features) and sample labels simultaneously from the actions/experiments. A hypothesis is tested based on the generated samples. By contrast, under our *label efficient* framework, we assume that the feature variables are already available, but the labels are costly. Hence, our work selects labels by accessing observed sample measurements. The literature more closely related to our approach is the experimental design literature such as [Simon and Simon, 2013, Bartroff and Lai, 2008, Lai et al., 2014, 2019] where a sample enrichment strategy is developed to enroll the patients responsive to an intervention to enlarge the intervention effect size. However, the sample enrichment strategy in [Simon and Simon, 2013, Bartroff and Lai, 2008, Lai et al., 2014, 2019] is designed for a two-sample mean difference test, and the test considered in our work is a two-sample independence test. Our work is also related to classifier two-sample tests [Lopez-Paz and Oquab, 2016]. A classifier two-sample test uses classifier accuracy to construct a two-sample testing statistic, and the trained classifier has the property that it can "explain which features are most important to distinguish distributions" [Lopez-Paz and Oquab, 2016]. We make use of this property of classifiers to devise the bimodal query scheme that is central to our approach. The devise query scheme is opposed to active learning work [Dasarathy et al., 2015, Li et al., 2020] that query labels near or on the decision boundaries.

## 2 PROBLEM STATEMENT

We consider a set of features and corresponding labels $\{(S_i, Z_i)\}_{i=1}^n \in \mathbb{R}^d \times \{0, 1\}$ i.i.d. generated from probability density function $p_{SZ}(S, Z)$. We write $\mathcal{S} = \{s_i\}_{i=1}^n$ to denote a set of observed features, and write $\mathcal{Z} = \{z_i\}_{i=1}^n$ to denote a set of observed labels corresponding to $\mathcal{S}$. We formally define null, $H_0$, and alternative, $H_1$, hypotheses as

$$H_0 : p_{S|Z}(\cdot \mid 0) = p_{S|Z}(\cdot \mid 1)$$
$$H_1 : p_{S|Z}(\cdot \mid 0) \neq p_{S|Z}(\cdot \mid 1). \tag{1}$$

Our novel problem formulation supposes that we have free access to the $s_i \in \mathcal{S}$, but that it is expensive to obtain the corresponding labels $z_i \in \mathcal{Z}$. We are however granted a label budget $n_q \leq n$, and we can select a size $n_q$ set $\bar{\mathcal{S}} \subseteq \mathcal{S}$ for which an oracle returns the corresponding label set $\bar{\mathcal{Z}} \subseteq \mathcal{Z}$. Notice that each $z_j \in \bar{\mathcal{Z}}$ is a sample from $p_{Z|S}(\cdot \mid s_j)$ where $s_j \in \bar{\mathcal{S}}$. The two-sample test considered in this paper aims to correctly reject $H_0$ in favor of $H_1$ using the samples in $\mathcal{S}$ and labels only for the samples $\bar{\mathcal{S}}$. Hereafter, we use $p(s|z)$, $p(z|s)$ and $p(s, z)$ as short forms of $p_{S|Z}(s \mid z)$, $p_{Z|S}(z \mid s)$ and $p_{SZ}(s, z)$. We similarly apply such abbreviations to other probability density functions introduced in other parts of the paper.

# 3 A FRAMEWORK FOR LABEL EFFICIENT TWO-SAMPLE HYPOTHESIS TESTING

In this section, we propose a three-stage framework for label efficient two-sample hypothesis testing. The corresponding algorithmic description is listed in Algorithm 1.

The inputs of the algorithm 1 are as follows: a feature set $\mathcal{S}$, a classification algorithm $\mathcal{A}$ that takes a training set as input and outputs a classifier, the number $n_t$ of labels used to construct a training set, the label budget $n_q$ and a pre-defined significance level $\alpha$. The output of algorithm 1 is a single bit of information: was the null hypothesis $H_0$ rejected? During the **first stage**, a classification algorithm $\mathcal{A}$ takes $n_t$ uniformly labeled samples (and corresponding labels provided by the oracle) as a training set input, and outputs a classifier with class probability estimation function $f : \mathbb{R}^d \rightarrow [0,1]$ used to model $p(Z = 1|s)$ subsequently. As classifiers such as neural networks and SVMs may be uncalibrated, a classifier calibration algorithm such as Platt scaling [Platt et al., 1999] could be incorporated into $\mathcal{A}$ to output a classifier with more accurate $f(s)$. We refer readers to [Platt et al., 1999] and [Niculescu-Mizil and Caruana, 2005] regarding the details of the calibration algorithm. During the **second stage**, we propose a bimodal query algorithm that queries the labels of samples with highest class one probability $f(s)$ and highest class zero probability $1 - f(s)$ until the label query budget, $n_q$, is exhausted. During the **third stage**, we split a labeled feature set $\bar{\mathcal{S}}$ to $\bar{\mathcal{X}}$ and $\bar{\mathcal{Y}}$, where each set only contains features from one class. Then the FR two-sample test is performed with the following steps: (1) compute the FR statistic (see section 4.1) from $\bar{\mathcal{X}}$ and $\bar{\mathcal{Y}}$; (2) compute $p$-value; (3) rejects the null hypothesis if the $p$-value is smaller than the pre-defined significance level $\alpha$.

# 4 THEORETICAL ANALYSIS OF THE THREE-STAGE FRAMEWORK

We begin by presenting the FR two-sample test [Friedman and Rafsky, 1979] in section 4.1, and then we frame label query as an optimization problem in section 4.2. From section 4.2.1 to section 4.4, we show that the solution to this optimization problem inspires the design of the three-stage framework, and the Type I error of the framework is controlled. In section 4.5, we discuss the extension of the proposed framework to using other two-sample tests.

## 4.1 THE FRIEDMAN-RAFSKY (FR) TWO-SAMPLE TEST

We consider paired feature and label samples $\{(s_i, z_i)\}_{i=1}^n \in \mathbb{R}^d \times \{0,1\}$ that are i.i.d realizations of $(S, Z) \sim p(s, z)$. We write $\mathcal{S} = \{s_1, \ldots, s_n\}$ to denote the

---

**Algorithm 1** A three-stage framework for the label efficient two-sample testing

**input** $\mathcal{S}, n_t, n_q, \alpha, \mathcal{A}$
**output** Reject or accept $H_0$
**First stage: model** $p(Z = 1|s)$
Uniformly sample $n_t$ features $\bar{\mathcal{S}} \subset \mathcal{S}$ and query their labels $\bar{\mathcal{Z}}$; $\mathcal{S} = \mathcal{S}/\bar{\mathcal{S}}$;
$\mathcal{A}$ takes input $\bar{\mathcal{S}}$ and $\bar{\mathcal{Z}}$, and outputs a classifier with class probability estimate function $f$ used to model $p(Z = 1 \mid s)$;
**Second stage: bimodal query**
Select $\lfloor (n_q - n_t)/2 \rfloor$ features $\bar{\mathcal{S}}_0 \subseteq \mathcal{S}$ which corresponds to $\lfloor (n_q - n_t)/2 \rfloor$ highest $f(s)$, and query their labels $\bar{\mathcal{Z}}_0$;
Select $n_q - n_t - \lfloor (n_q - n_t)/2 \rfloor$ features $\bar{\mathcal{S}}_1 \subseteq \mathcal{S}$ which corresponds to $n_q - n_t - \lfloor (n_q - n_t)/2 \rfloor$ highest $1 - f(s)$, and query their labels $\bar{\mathcal{Z}}_1$;
$\bar{\mathcal{S}} = \bar{\mathcal{S}} \bigcup \bar{\mathcal{S}}_0 \bigcup \bar{\mathcal{S}}_1$; $\bar{\mathcal{Z}} = \bar{\mathcal{Z}} \bigcup \bar{\mathcal{Z}}_0 \bigcup \bar{\mathcal{Z}}_1$
**Third stage: FR two-sample test**
Split $\bar{\mathcal{S}}$ to two groups $\bar{\mathcal{X}}$ and $\bar{\mathcal{Y}}$ based on the label set $\bar{\mathcal{Z}}$; compute FR statistic using $\bar{\mathcal{X}}$ and $\bar{\mathcal{Y}}$; compute $p$-value;
**If** $p < \alpha$ **Then** Reject $H_0$ **Else** Accept $H_0$.

---

set of feature observations and write $\mathcal{Z} = \{z_1, \ldots, z_n\}$ to denote the set of corresponding label observations. Furthermore, we divide $\mathcal{S}$ in two sets based on the label $z_i$ of $s_i$, and get $\mathcal{X} = \{x_1, \ldots, x_{n_0}\}$ from class zero and $\mathcal{Y} = \{y_1, \ldots, y_{n_1}\}$ from class one where $\mathcal{S} = \mathcal{X} \bigcup \mathcal{Y}$ and $n = n_0 + n_1$. Friedman and Rafsky [1979] proposed a non-parametric two-sample test statistic that is computed as follows: First, one constructs a Euclidean minimum spanning tree (MST) over the samples $\mathcal{X}$ and $\mathcal{Y}$, i.e., the MST of a complete graph whose vertices are the samples, and edge weights are the Euclidean distance between the samples. Then, one counts the edges connecting samples from opposite classes (i.e., cut edges). We use $r_n$ to denote the cut-edge number for the MST constructed over $\mathcal{S}$; $r_n$ corresponds to an observation of the corresponding random variable $R_n$ that models the cut-edge number for an MST constructed from $\{S_i, Z_i\}_{i=1}^n$. Under the alternative hypothesis $H_1$, $r_n$ is expected to be small, and under the null hypothesis $H_0$, $r_n$ is expected to be large. The Friedman-Rafsky (FR) test statistic $w_n$ is a normalized version of $r_n$,

$$w_n = \frac{r_n - \mathrm{E}[R_n \mid H_0, \mathcal{S}]}{\sqrt{\mathrm{Var}[R_n \mid H_0, \mathcal{S}]}}, \quad (2)$$

where $\mathrm{E}[R_n \mid H_0, \mathcal{S}]$ and $\mathrm{Var}[R_n \mid H_0, \mathcal{S}]$ are the expectation and the variance of $R_n$ conditional on $\mathcal{S}$ under the null hypothesis $H_0$. We use $W_n$ to denote a random variable of which $w_n$ is a realization. $W_n$ is a random FR statistic obtained from $n$ i.i.d pairs of $\{S_i, Z_i\}_{i=1}^n \sim p(s, z)$. Since $r_n$ is the number of the cut-edges connecting opposite labels, calculating $r_n$ requires knowledge of both $\mathcal{S}$ and $\mathcal{Z}$. On the other hand, the derivation for $\mathrm{E}[R_n \mid H_0, \mathcal{S}]$ and $\mathrm{Var}[R \mid H_0, \mathcal{S}]$ under $H_0$ are label free due to the inde-

pendency between $Z$ and $S$. The numerical expression of $E[R_n \mid H_0, \mathcal{S}]$ and $\text{Var}[R_n \mid H_0, \mathcal{S}]$ can be found in appendix. The FR test rejects $H_0$ if a small $W_n$ is observed.

In practice as stated in [Friedman and Rafsky, 1979], the FR test is carried out as a permutation test where the null distribution (distribution of a statistic under the null $H_0$) of $W_n$ is obtained by calculating all possible values of $w_n$ (2) under all possible rearrangements of the observations of $\mathcal{S}$. Then a $p$-value is obtained using the permutation null distribution and the $w_n$ computed from $\mathcal{X}$ and $\mathcal{Y}$. The $p$-value is compared to a significance level $\alpha$ to reject $H_0$ for $p < \alpha$. We refer readers to [Welch, 1990] for the procedure of the permutation test. Both Theorem 4.1.2 in [Bloemena, 1964] and Section 4 in [Friedman and Rafsky, 1979] demonstrate that, if $W_n$ is generated under $H_0$, then the *permutation distribution* of $W_n$ approaches a standard normal distribution for large sample size $n \to \infty$: $W_n \xrightarrow{\mathcal{D}} \mathcal{N}(0, 1)$, where $\xrightarrow{\mathcal{D}}$ stands for distributional convergence. Therefore, we follow [Friedman and Rafsky, 1979] and use $\mathcal{N}(0, 1)$ as the null distribution of $W_n$, and we get the $p$-value given by

$$p = \phi[W_n], \qquad (3)$$

where $\phi$ is the cumulative function of the standard normal distribution. We use $P_i(E)$ to denote the probability of an event under $H_i$. Two types of error for a two-sample test are considered: the Type I error $P_0(p < \alpha)$ rejects $H_0$ when $H_0$ is true, and the Type II error $1 - P_1(p < \alpha)$ rejects $H_1$ when $H_1$ is true. $P_1(p < \alpha)$ is called the power of the test.

The authors in [Henze and Penrose, 1999] further show an asymptotic property of the FR testing statistic $W_n$, and we restate (an equivalent version of) their results in the following. This restated result will be useful in section 4.2.1 to show that the proposed bimodal query is inspired by the asymptotic minimization of $W_n$. Following [Henze and Penrose, 1999], we suppose that there is a constant $u \in [0, 1]$ such that as $n$ tends to infinity, $n_0/n \to u$; this is known as the *usual limiting regime*. Note that $u$ can be thought of as the class prior probability for $Z = 0$ and we write $v = 1 - u$ to denote the class prior probability for $Z = 1$. Under the *usual limiting regime*, combining Theorem 2 in [Henze and Penrose, 1999] and Theorem 3 in [Steele et al., 1987] yields an almost sure result for $\frac{W_n}{n}$:

**Theorem 1.** *Under the usual limiting regime,*

$$\lim_{n \to \infty} \frac{W_n}{n} = \frac{[\int 2p(Z = 0 \mid s)p(Z = 1 \mid s)p(s)ds - 2uv]}{\sqrt{2uv[2uv + (A_d - 1)(1 - 4uv)]}}$$
$$(4)$$

*almost surely, where $A_d$ is a constant dependent on the dimension $d$.*

We refer the readers to appendix for the proof. Briefly, Theorem 1 results from combining three almost sure convergence results: $\frac{R_n}{n} \to \int 2p(Z = 0 \mid s)p(Z = 1 \mid s)p(s)ds$,

$\frac{\text{E}[R_n \mid H_0, \{S_i\}_{i=1}^n]}{n} \to 2uv$ and $\text{Var}[R_n \mid H_0, \{S_i\}_{i=1}^n] \to \sqrt{2uv[2uv + (A_d - 1)(1 - 4uv)]}$ for $n \to \infty$.

## 4.2 A LABELING SCHEME THAT MINIMIZES THE FR STATISTIC $W_n$

Our problem statement assumes that the feature set $\mathcal{S} = \{s_1, \dots, s_n\}$ and the label set $\mathcal{Z} = \{z_1, \dots, z_n\}$ are i.i.d realizations of $(S, Z) \sim p(s, z)$, and that the access to every $s_i \in \mathcal{S}$ is free; but it is costly to obtain the corresponding label $z_i \in \mathcal{Z}$. However, we are assigned a label budget $n_q$ such that we can select a set $\bar{\mathcal{S}} \subseteq \mathcal{S}$ to query labels from an oracle, and each random variable $Z_i$ corresponding to the returned label $z_i$ admits $p(z|s_i)$. We then divide $\bar{\mathcal{S}}$ to $\bar{\mathcal{X}}$ from class zero and $\bar{\mathcal{Y}}$ from class one and perform a two-sample test on $\bar{\mathcal{X}}$ and $\bar{\mathcal{Y}}$. We write $|\bar{\mathcal{X}}| = \bar{n}_0$ and $|\bar{\mathcal{Y}}| = \bar{n}_1$ and we have $n_q = \bar{n}_0 + \bar{n}_1$.

Our aim is to find a query scheme that increases the testing power of a test performed on the selected samples $\bar{\mathcal{X}}$ and $\bar{\mathcal{Y}}$. For a uniform sampling query scheme, then we will have $\bar{\mathcal{S}}$ as a set of $n_q$ i.i.d realizations generated from the original marginal distribution $p(s)$, and we can rewrite $p$-value in (3) as $p = \phi[W_{n_q}]$ where $W_{n_q}$ is a FR statistic random variable obtained from $n_q$ i.i.d pairs of $(S_i, Z_i) \sim p(s, z)$. Instead of directly tackling the query scheme, we consider to find an optimal marginal distribution $p_{q^*}(s)$ such that, under the alternative hypothesis $H_1$, performing the FR test on a set of i.i.d. $S_i \sim p_{q^*}(s)$ generates large testing power than performing on the uniformly sampled data points with the same number of labels $n_q$. After identifying the optimal marginal $p_{q^*}(s)$, in practice we will use a query scheme to find a set of features $\bar{\mathcal{S}} \subseteq \mathcal{S}$ similar to $n_q$ i.i.d realization of $S_i \sim p_{q^*}(s)$. This motivates the bimodal query scheme in algorithm 1 to increase the power of the FR test.

### 4.2.1 A marginal distribution to minimize the FR statistic asymptotically

Given $n_q$ i.i.d. realizations generated from $p_q(s)$, we seek a $p_q(s)$ to minimize $W_{n_q}$ and hence generate a more powerful FR test. From Theorem 1 we know that the convergence result of $\frac{W_{n_q}}{n_q}$ is a function of only $p_q(s)$ under the usual limiting regime $\frac{\bar{n}_0}{n_q} \to u$ and $\frac{\bar{n}_1}{n_q} \to v$. Therefore, we construct the following optimization problem:

$$\min_{p_q(s)} \int p(Z = 0 \mid s)p(Z = 1 \mid s)p_q(s)ds$$

$$\text{subject to } \int p(Z = 0 \mid s)p_q(s)ds = u$$

$$\int p_q(s)ds = 1, \quad p_q(s) \geq 0. \qquad (5)$$

Under the null hypothesis $H_0$, $Z$ and $S$ are independent and thus $p(s, z) = p(s)p(z)$, and $\int p(Z = 0|s)p(Z = 1|s)p_q(s)ds = uv$ for any $p_q(s)$. Therefore, minimizing 5 with $p_q(s)$ does not alter the Type I error. A more thorough

analysis of the Type I error is provided in section 4.4. On the other hand, under the alternate $H_1$, solving the optimization problem (5) leads to a solution that minimizes $W_{n_q}$ in 3 for large sample sizes $n_q \to \infty$, leading to a decreasing Type II error of the FR test.

We approximate the continuous random variable $S$ in Eq. (5) with a discrete versions of the same by partitioning the support of $p_q(s)$ into balls $B(s_i, r) \subseteq \mathbb{R}^n$ with radius $r$ centering at $s_i$ which leads to discrete $p(Z = 0|s_i) = \int_{B(s_i,r)} p(Z = 0|s)p(s)ds$. This converts the optimization problem to a linear program (6)

$$\max_{p_q(s_i)} \sum_i p(Z = 0 \mid s_i)^2 p_q(s_i)$$

$$\text{subject to} \sum_i p(Z = 0 \mid s_i)p_q(s_i) = u$$

$$\sum_i p_q(s_i) = 1, \quad p_q(s_i) \geq 0. \tag{6}$$

Note that $p(Z = 1|s)$ in Eq. (5) is replaced by $1 - p(Z = 0|s)$ and optimization problem is modified accordingly.

**Theorem 2.** *The optimal solution $p_{q^*}(s_i)$ to the LP in (6) is,*

$$p_{q^*}(s_{q_0}) = \frac{u - p(Z = 0 \mid s_{q_1})}{p(Z = 0 \mid s_{q_0}) - p(Z = 0 \mid s_{q_1})},$$

$$p_{q^*}(s_{q_1}) = \frac{p(Z = 0 \mid s_{q_0}) - u}{p(Z = 0 \mid s_{q_0}) - p(Z = 0 \mid s_{q_1})},$$

$$p_{q^*}(s_i) = 0 \quad \forall i \notin \{q_0, q_1\}$$
*where $q_0 = \arg\min_i[p(Z = 0 \mid s_i)] = \arg\max_i[p(Z = 1 \mid s_i)]$,*

$$q_1 = \arg\max_i[p(Z = 0 \mid s_i)]. \tag{7}$$

Briefly, the derivation of Eq. (7) comes about when we combine the linear constraints in Eq. (6) with the fact that the optimum value is always achieved on the boundary of the constraint set for LP problems [Korte et al., 2011]. We refer readers to appendix for details. The optimal solution $p_{q^*}(s_i)$ of Eq. (6) is a bimodal delta function (with modes at $q_0$ and $q_1$) that samples the highest posterior probabilities of $p(Z = 0|s_i)$ and $p(Z = 1|s_i)$. Reducing the radius $r$ of a ball $B(s_i, r)$ towards zero makes $p_q(s_i)$ a nearly probability density function therefore the derived $p_{q^*}(s_i)$ in (7) is regarded as an optimal solution to minimize the original objective function (5).

#### 4.2.2 Practicality of the proposed framework

Theorem 2 tells us that drawing $n_q$ i.i.d. samples from $p_{q^*}(s)$ to label is an ideal query scheme to increase the testing power of the FR test. However, practical utility of $p_{q^*}(s)$ (7) to minimize $W_{n_q}$ is complicated by two facts: (1) $p(z|s)$ is

unknown to us, and (2) we do not have a random sample generator to generate $n_q$ i.i.d. samples from $p_q(s)$. In practice, we approximate $p(z|s)$ by the output probability of a classifier and symmetrically query the labels of points at the approximated highest $p(Z = 0|s)$ and $p(Z = 1|s)$. This motivates the use of a classifier during the first stage for driving the bimodal query labeling scheme during the second stage. The idea to use a probabilistic classifier to estimate $p(z|s)$ has been similarly used in many previous works [Friedman, 2004, Lopez-Paz and Oquab, 2016, Kossen et al., 2021]. We include extensive experimental results using different classifiers in appendix.

With respect to the second point, we empirically demonstrate that selecting features by bimodal query increases the power of the test across several applications; all while controlling the Type I (see section 4.4) even given non-i.i.d. features.

### 4.3 CONVERGENCE OF AN EXPECTED FR STATISTIC VARIANT

The cost function in Eq. (5) is motivated by the almost sure results outlined in Theorem 1. In this section, we consider a FR statistic variant and show that the expected FR statistic variant converges in $\mathcal{O}(n_q^{-1})$ ($n_q$ is label budget) for $n_q$ features i.i.d. generated from the bimodal delta function $p_{q^*}(s)$ in 7, and the convergence rate $\mathcal{O}(n_q^{-1})$ is independent of feature dimension $d$.

Given a FR-test performed on $n$ samples $\{(S_i, Z_i)\}_{i=1}^n$ i.i.d generated from a marginal distribution $p(s, z)$, we have the expectation of the FR statistic in (2) as $\mathrm{E}[W_n] = \mathrm{E}\left[\frac{R_n - \mathrm{E}[R_n|H_0, \{S_i\}_{i=1}^n]}{\sqrt{\mathrm{Var}[R_n|H_0, \{S_i\}_{i=1}^n]}}\right]$. In this subsection, we use $\mathcal{X}$ and $\mathcal{Y}$ to denote sets of feature random variables $S_i$ with membership $Z_i = 0$ and $Z_i = 1$ respectively. The expected $R_n$ under the null $H_0$ is only determined by size $n_0 = |\mathcal{X}|$ and size $n_1 = |\mathcal{Y}|$ (see appendix), which leads to $\mathrm{E}[R_n|H_0, \{S_i\}_{i=1}^n] = \mathrm{E}[R_n|H_0, |\mathcal{X}|, |\mathcal{Y}|]$. However, the variance $\mathrm{Var}[R_n|H_0, \{S_i\}_{i=1}^n]$ under the null hypothesis is dependent on the topology of MST constructed over $\{S_i\}_{i=1}^n$ and is intractable. This makes the evaluation of $\mathrm{E}[W_n]$ difficult. Therefore, following [Henze and Penrose, 1999], we decouple $\mathrm{Var}[R_n|H_0, \{S_i\}_{i=1}^n]$ from $W_n$ in Eq. (2) by multiplying $\sqrt{\mathrm{Var}[R_n|H_0, \{S_i\}_{i=1}^n]}$ and generate a variant of the FR statistic random variable, $\overline{W}_n = R_n - \mathrm{E}[R_n|H_0, |\mathcal{X}|, |\mathcal{Y}|]$. In what follows, we evaluate the expected FR statistic variant $\mathrm{E}[\overline{W}_{n_q}] = \mathrm{E}[R_{n_q}] - \mathrm{E}[R_{n_q}|H_0]$ given $n_q$ features $S_i$ i.i.d. generated from $p_{q^*}(s)$. Specifically, we evaluate $\mathrm{E}\left[\frac{\overline{W}_{n_q}}{n_q}\right] = \mathrm{E}\left[\frac{R_{n_q}}{n_q}\right] - \mathrm{E}\left[\frac{R_{n_q}}{n_q} \mid H_0\right]$ and state the following theorem.

**Theorem 3.** *Given that $n_q$ samples are i.i.d. generated from*

$p_{q^*}(s)$ (7), we have

$$\mathrm{E}\left[\frac{\overline{W}_{n_q}}{n_q}\right] = \int 2p(Z=0|s)p(Z=1|s)p_{q^*}(s)ds$$
$$+ \mathcal{O}(n_q^{-1}) - 2uv \tag{8}$$

The difficulty in evaluating $\mathrm{E}\left[\frac{\overline{W}_{n_q}}{n_q}\right]$ comes from the evaluation of $\mathrm{E}\left[\frac{R_{n_q}}{n_q}\right]$. Fortunately, considering $p_{q^*}(s)$ (7) is a discrete marginal distribution with two modes at $s_{q_0}$ and $s_{q_1}$ ($q_0 = \arg\max_i[p(Z=1|s_i)]$ and $q_1 = \arg\max_i[p(Z=0|s_i)]$, see (7)) and the probabilities at other points are zero, we can precisely obtain the probability of an edge being a cut-edge at $s_{q_0}$ or $s_{q_1}$ thereby leading to convenient evaluation of $\mathrm{E}\left[\frac{R_{n_q}}{n_q}\right]$. We refer readers to appendix for the proof.

*Remark* 1. For the original FR test (or equivalent to our framework with the bimodal query replaced by the uniform sampling), given sample size $n_q$, the expected FR variant $\mathrm{E}\left[\overline{W}_{n_q}\right]$ inflates with increasing dimension $d$ and hinders differentiating the alternative hypothesis from the null hypothesis. Using $p_{q^*}(s)$ (7) turns out to not only minimize the convergence of $\frac{W_{n_q}}{n_q}$ (4), but also results in a convergence rate of $\mathcal{O}(n_q^{-1})$ for $\mathrm{E}\left[\frac{\overline{W}_{n_q}}{n_q}\right]$. This convergence rate is independent of dimension $d$; therefore, performing a FR test on samples generated from $p_{q^*}(s)$ can effectively suppresses the inflation of $\mathrm{E}[\overline{W}_{n_q}]$ for high-dimension samples and helps reject the null under the alternative hypothesis.

### 4.4 TYPE I ERROR OF THE THREE-STAGE FRAMEWORK

One important observation for the proposed framework is that the features labeled in the second stage are dependent on the uniform sampled features in the first stage. For every $n$ i.i.d. realizations $\{s_i, z_i\}_{i=1}^n$ of $\{S_i, Z_i\}_{i=1}^n \sim p(s, z)$ under the null hypothesis $H_0$, we write $\bar{S} = \{\bar{s}_1, \ldots, \bar{s}_{n_q}\}$ to denote a set that our query scheme (comprised of uniform sampling and bimodal query) selects from $S = \{s_1, \ldots, s_n\}$, and write $\bar{Z} = \{\bar{z}_1, \ldots, \bar{z}_{n_q}\}$ to denote a set of label observations corresponding to $\bar{S}$. We use $\bar{S}_i$ and $\bar{Z}_i$ to denote the random variables corresponding to $\bar{s}_i$ and $\bar{z}_i$. Under the $H_0$, or equivalently, $S \perp Z$, an improper use of the bimodal query might tend to label samples in the region with high bias, and makes $\bar{S}_i$ dependent on $\bar{Z}_i$, and hence increase the Type I error. In the following, we present our theorem regarding the Type I error control:

**Theorem 4.** *Suppose $(\bar{S}_i, \bar{Z}_i)_{i=1}^{n_q}$ are pairs of random feature variables and label variables acquired in the end of the second stage of the framework, using a permutation test in the third stage of the framework to obtain $p$-value from $(\bar{S}_i, \bar{Z}_i)_{i=1}^{n_q}$ for any two-sample test have Type I error $P(p \le \alpha) \le \alpha, \forall \alpha$ for the framework.*

Theorem 4 states that the Type I error of our framework is upper-bounded by $\alpha$ for any two-sample test carried out as a permutation test in the third stage. A permutation test rearranges labels of features, obtains permutation distribution of a statistic computed from the rearrangements, and rejects $H_0$ if a true observed statistic is contained in $\alpha$ probability range of the permutation distribution. This process does not need features to be i.i.d. sampled to control the Type I error at exact $\alpha$, and it is applicable to any two-sample tests testing independency between $\bar{S}_i$ and $\bar{Z}_i$. However, we need to make sure our query procedure maintains $\bar{S}_i \perp \bar{Z}_i$ under the $H_0$. Our framework only trains a classifier one time with uniformly sampled data points in the first stage, and then the bimodal query selects a subset of features from $S$ to label based on the trained classifier. For a set of feature and label variables $\mathcal{Q} = \{\bar{S}_i, \bar{Z}_i\}_{i=1}^{n_q}$ obtained in the end of the second stage, we write $\mathcal{Q}_u \subseteq \mathcal{Q}$ to denote the set obtained from uniform sampling, and write $\mathcal{Q}_b \subseteq \mathcal{Q}$ to denote the set obtained from bimodal query. Considering that a uniform sampling scheme does not change the original distributional properties ($S \perp Z$ under the null) to generate $(\bar{S}_i, \bar{Z}_i) \in \mathcal{Q}_u$, we have $\bar{S}_i \perp \bar{Z}_i, \forall(\bar{S}_i, \bar{Z}_i) \in \mathcal{Q}_u$. $\mathcal{Q}_b$ is not used to train the classifier, so we also have $\bar{S}_i \perp \bar{Z}_i, \forall(\bar{S}_i, \bar{Z}_i) \in \mathcal{Q}_b$. We refer readers to appendix for details.

### 4.5 EXTENSIBILITY OF THE THREE-STAGE FRAMEWORK

The starting point for developing the bimodal query used in the proposed framework is Theorem 1. This asymptotic result appears in many graph-based two-sample tests where the testing statistic is a function of cut-edge number [Chen and Friedman, 2017, Rosenbaum, 2005, Schilling, 1986, Henze, 1988, Chen et al., 2018]. Furthermore, the Theorem 4 states that our framework controls Type I error for any two-sample tests if a permutation test is used. The above two reasons guarantee that, when replacing FR test with other two-sample tests in the third stage, the Type I error is controlled if a permutation test is used, and the bimodal query is a reasonable rule for increasing the testing power of a test. In the experimental results, we empirically demonstrate the extensibility of our framework by using the Chen test [Chen and Friedman, 2017] and the cross-matching test [Rosenbaum, 2005].

## 5 EXPERIMENTAL RESULTS

The proposed framework attributes the increasing testing power of the FR test for a label budget to the use of the bimodal query in the second stage. We therefore replace the

bimodal query with **passive query**, **uncertainty query** and **certainty query** to establish three baselines. The passive query uniformly samples datapoints to query. The uncertainty query selects the points at the smallest $p(z|s)$ (the most uncertain point). The certainty query scheme is a heuristic that select points at the most certain region–highest $p(z|s)$. We also extend the framework beyond FR test to using the Chen test [Chen and Friedman, 2017] and the cross-matching test [Rosenbaum, 2005] to empirically investigate the extensibility of the proposed framework to other two-sample tests. The three two-sample tests all have known asymptotic or exact permutation null distributions.

## 5.1 EXPERIMENTS ON SYNTHETIC DATASETS

**Data generated under** $H_1$ **being true**: we use a two-dimensional normal distribution to generate two types of binary-class synthetic datasets with a sample size of 2000. One type has the data with two groups generated from $\mathcal{N}((\delta_1, 0), I_2)$ and $\mathcal{N}((-\delta_1, 0), I_2)$, and the other type has data with two groups generated from $\mathcal{N}((\delta_2, 0), I_2)$ and $\mathcal{N}((-\delta_2, 0), I_2(1 + \sigma))$. We set $\delta_1 = 1$, $\delta_2 = 0.6$ and $\sigma = 1$. The two different ways to generate data result in a location alternative $H_1^1$(mean difference) and scale alternative $H_1^2$(variance difference) for the two-sample hypothesis test to detect. Both types of data are considered as the data realizations of different distributions which implies $H_0$ should be rejected.

**Data generated under** $H_0$ **being true**: we simply generate two groups of data both from same distribution $\mathcal{N}(\mathbf{0}, I_2)$.

We repeat the above procedure 200 times to generate enough cases for a fair performance evaluation. We remove the labels of the synthetic dataset and use the three-stage framework shown in the algorithm 1 to perform label-efficient two-sample hypothesis testing. We set $n_t = 50$ and use logistic regression as the classification algorithm input $\mathcal{A}$. We set $\alpha = 0.05$, and set $n_q$ from 10% to 100% of the whole data size to evaluate the performance of the proposed framework and the three baselines. In addition to the FR test [Friedman and Rafsky, 1979] proposed to used in the framework, Chen test [Chen and Friedman, 2017] and cross-match test [Rosenbaum, 2005] are also used to examine the extensibility of the framework to using other two-sample tests. A promising framework should control the Type I error (upper-bounded by $\alpha = 0.05$) under the null $H_0$ and decrease the Type II error under the alternative hypothesis $H_1$.

Figure 2(a) shows the Type II errors returned by the proposed framework and its parallel implementations with the bimodal query replaced by the three baseline queries. It is observed that the proposed framework generates lower Type II error than its parallel implementation with only a small label proportions of the whole datasize. Figure 3(a) shows the 95% confidence of the Type I error returned by

the proposed framework. It is observed that the significance level $\alpha = 0.05$ overlaps with the 95% confidence interval of the Type I error, which agrees with the Theorem 4 that the Type I error of the proposed framework is upper-bounded by $\alpha$. We refer readers to appendix for the results of the Chen test [Chen and Friedman, 2017] and the cross-match test [Rosenbaum, 2005] and the results of using other classification algorithms, which shows the extensibility of the proposed framework to using other two-sample tests and other classification algorithms.

## 5.2 EXPERIMENTS ON MNIST AND ADNI

**MNIST data generated under** $H_1$: we sample images from MNIST [LeCun, 1998] to create two groups of data as follows: in the group one, we randomly sample 1000 images of one class from MNIST; and in the group two, we first randomly sample 700 images of the same class but sample the other 300 images of a different class from the MNIST. Both groups are projected to a 28-dimensional space by a convolutional autoencoder [Ng et al., 2011] before injecting to the proposed three-stage framework. The second group of data should follow a distribution similar to the group one however it is polluted by a different class of data. We repeat the above data generating process 200 times and ideally a two-sample test should reject the null hypothesis $H_0$ for each case.

**MNIST data generated under** $H_0$: we simply sample two groups of 1000 images from one class in the MNIST data. We repeat the above process 200 times to obtain 200 cases of MNIST data under $H_0$.

**The Alzheimer's Disease Neuroimaging Initiative (ADNI) dataset**: data from the Alzheimer's Disease Neuroimaging Initiative (ADNI) database [Jack Jr et al., 2008] was obtained to demonstrate a real-world application of the label efficient two-sample testing. Our ADNI dataset is comprised of five cognition measurement scores obtained from participants in ADNI. In addition, ADNI has an available PET-imaging based measure used to quantify amyloid load (AV45) in the brains of patients with AD patients [Gruchot et al., 2011]. This motivates a hypothesis that the five measures are different in individuals with amyloid in the brain from those without amyloid in the brain. That is, $H_0$ implies that the five cognition measurement scores from participants with high or low AV45 have no significant and $H_1$ implies the opposite. Measuring AV45 requires a PET scan, a costly procedure that we would like to minimize. Therefore we use the proposed framework to perform a two-sample test with fewer PET scans (label queries). In the experiment, we binarize the AV45 using the cut-off value suggested by ADNI. We sample 750 participants with AV45 values higher than the cut-off as group one, and sample 250 participants with AV45 values lower than the cut-off as group two. We repeat the above random sampling 200 times to generate 200 data cases.

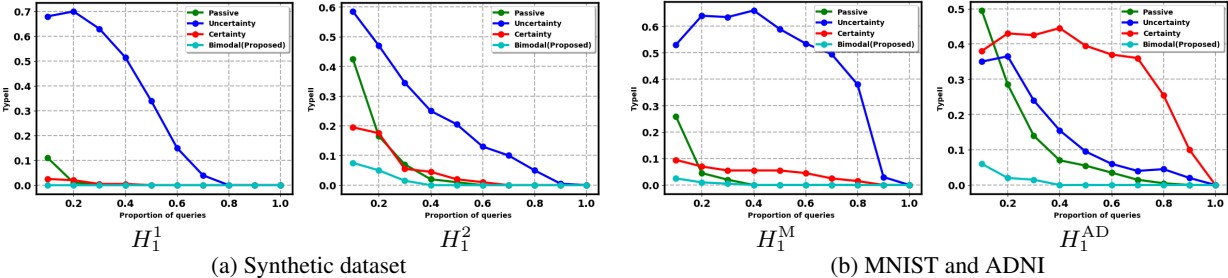

(a) Synthetic dataset      (b) MNIST and ADNI

Figure 2: Type II error of the proposed framework and its parallel implementations with the bimodal query replaced with three baseline queries under the two synthetic dataset alternative hypotheses $H_1^1$ and $H_1^2$ and under the MNIST and ADNI alternative hypotheses $H_1^M$ and $H_1^{AD}$. Type II error is on the Y-axis and label budget (percentage of all data) is on the X-axis.

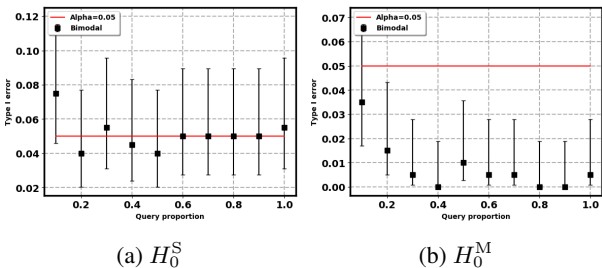

(a) $H_0^S$      (b) $H_0^M$

Figure 3: Type I error (95% confidence interval) of the proposed framework under the synthetic and MNIST null hypotheses $H_0^S$ and $H_0^M$. Type I error is on the Y-axis and label budget (percentage of all data) is on the X-axis.

For the MNIST dataset, we set $n_t = 100$ and vary $n_q$ from 10% to 100% of the whole dataset with 10% interval increment. We use a neural network to model $p(z|s)$. For the ADNI dataset, we set $n_t = 50$ and also vary $n_q$ from 10% to 100%. We use logistic regression to model $p(z|s)$. We set $\alpha = 0.05$ for both cases.

We compare the proposed framework to its parallel implementations to demonstrate the increased testing power of the bimodal query-based FR test relative to the baseline query-based FR tests. This can be seen in Figure 2(b) where the proposed framework generates lower Type II error in both MNIST and ADNI than its parallel implementations with only a small label query proportion of the whole dataset size. Then in Figure 3(a), we observe that the significance level $\alpha = 0.05$ either overlaps with or upper-bounds the 95% confidence interval of the Type I error of the proposed framework. Both results above demonstrate that the proposed framework increases the testing power with same label budget $n_q$ and also can control the Type I error for real datasets. Lastly, we replace the FR test in the framework with the Chen test [Chen and Friedman, 2017] and the cross-match test [Rosenbaum, 2005] to examine the extensibility of the proposed framework to using other two-sample tests for the real datasets. We refer readers to appendix for the results of the Chen test [Chen and Friedman, 2017] and the cross-match test [Rosenbaum, 2005] and the results of using other classification algorithms. We observe that our framework with the FR test replaced by the Chen and the

cross-match tests still return lower Type II errors than the parallels using other baseline queries with a small label query proportion, while controlling the Type I error at a desired level.

## 5.3 ABLATION STUDY ON THEOREM 3

In this section, we study the Theorem 3 that alludes the testing power of the proposed framework is dimension free. We reuse the data generation paradigm under the $H_1^1$ in section 5.1 but increase the dimension number $d$ from 2 to 18, and therefore create 200 data cases having two groups of samples gener-

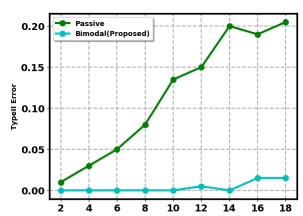

Figure 4: Type II error returned by the proposed framework and its parallel of the uniform sampling based FR test for various dimension $d$.

ated from $\mathcal{N}((\delta_1, ..., 0)^d, I_d)$ and $\mathcal{N}((-\delta_1, ..., 0)^d, I_d)$ for $d = 2, ..., 18$ and $\delta_1 = 1$. We then use the proposed framework and its parallel of uniform sampling-based FR test to test the generated high-dimensional dataset under the alternate $H_1^1$. We set $n_q = 20\%$ of the whole datasize. Figure 4 shows that the Type II error of the proposed framework does not vary much for different dimensions but the Type II error of the passive query based FR test explodes along the increasing sample dimension. This empirical observation is consistent with the results of Theorem 3.

## 6 CONCLUSION

We extend the traditional two-sample hypothesis testing to a new important setting where the sample measurements are available but the group labels are unknown and costly to obtain. We therefore devise a three-stage framework for the label efficient two-sample test based on theoretical foundations of increasing the testing power and controlling the Type I error with a label budget.

## ACKNOWLEDGEMENT

This work was funded in part by Office of Naval Research grant N00014-21-1-2615 and by the National Science Foundation (NSF) under grants CNS-2003111, CCF-2007688, and CCF-2048223.

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
