# OpenReview forum: "A label-efficient two-sample test"
_auai.org/UAI/2022/Conference — UAI 2022 Poster_

### Official Review · Reviewer_RGHh · 2022-04-10

**Q2(1) Originality/Novelty:** 3
**Q2(2) Significance/Impact:** 3
**Q2(3) Correctness/Technical Quality:** 3
**Q2(6) Clarity Of Writing:** 4
**Q6 Overall Score:** 6
**Q8 Confidence In Your Score:** 4

**Q1 Summary And Contributions:**

This paper proposes a new active setting for the typical two-sample test, where the labels are costly. This paper designs an algorithm guided by minimizing the FR statistic asymptotically to solve this problem. By adjusting the distribution where features to be queried are sampled, the proposed algorithm can control the Type-I error under the null hypothesis and decrease the Type-II error under the alternative hypothesis.

**Q2 Assessment Of The Paper:**

More detailed information regarding each of these aspects is given below:

**Q2(4) Quality Of Experiments (Optional):**

3: Good: The experimental evaluation is adequate, and the results convincingly support the main claims.

**Q2(5) Reproducibility:**

4: Excellent: Key resources (e.g., proofs, code, data) are available and key details (e.g., proof sketches, experimental setup) are comprehensively described for competent researchers to confidently and easily reproduce the main results.

**Q3 Main Strengths:**

- The setting is novel and practical. This paper considers a variant of the classical two-sample setting, where labels are queried actively. This setting can be encountered in realistic scenarios where labels are costly to be obtained.
- The algorithm is guided by theoretical analysis. This paper tries to decrease the Type-II error by solving a minimization problem about the FR statistic $W_n$, making the algorithm non-heuristic.
- The theoretical results are solid and well presented.
- The experimental results are convincing and support the major claims.


**Q4 Main Weakness:**

The theoretical results do not exactly match the problem setting and algorithm design:
- Theorem 1 requires a large sample size ($n_q \rightarrow \infty $), whereas this paper mainly focuses on a label-efficient scenario, which means $n_q$ is small. Can the theoretical results truly guide the algorithm design?
- The meaning of notation $n_q$ in Algorithm 1 and Theorem 3 is different. For Algorithm 1, $n_t$ of $n_q$ samples are drawn uniformly and $n_q – n_t$ of $n_q$ samples are drawn from the so-called optimal distribution $p_{q^{\star}}(s)$. However, for Theorem 3, all of the $n_q$ samples are i.i.d. drawn from $p_{q^{\star}}(s)$.
- There is no discussion about $n_t$. How many samples need to be allocated for training the classifier? Whether the accuracy of the posterior probability estimation affects the algorithm performance?  It seems that efficient labels may cause a poor generation error of the classifier.


**Q5 Detailed Comments To The Authors:**

- In the experiments, only 100 samples of the MNIST images are used to train the neural network. Some samples may be needed for validation if calibration is conducted. Are the samples enough to train a well-performed classifier? Maybe the accuracy of the classifier or the estimation error of the posterior probabilities should be reported.
- It seems that $p_{q^{\star}}(s_{ q_{0}})$ is not always equal to $p_{q^{\star}}(s_{ q_{1}})$ in Theorem 2, but why does Algorithm 1 selects the same number of high $f(s)$ and low $f(s)$? If the selected ratio ($|\bar{S_0}|/|\bar{S_1}|$) is changed, will the performance be affected?


**Q7 Justification For Your Score:**

This paper studies a novel and practical problem and proposes a feasible algorithm that is better than the baselines. Although the theoretical results and algorithms do not match in some aspects, the theoretical results actually motivate the design of the algorithms. Overall, I find the merit overrides the flaw of this paper and would like to weak accept this paper.

**Q9 Complying With Reviewing Instructions:**

1: Yes.

---

### Official Review · Reviewer_4Fb2 · 2022-04-12

**Q2(1) Originality/Novelty:** 4
**Q2(2) Significance/Impact:** 3
**Q2(3) Correctness/Technical Quality:** 3
**Q2(6) Clarity Of Writing:** 3
**Q6 Overall Score:** 8
**Q8 Confidence In Your Score:** 3

**Q1 Summary And Contributions:**

The paper introduces an extension of the two-sample test to unlabelled data. The authors address situations where there is a need to compare to groups of samples, but determining the membership in each group is costly (e.g. it involves running expensive lab tests). The authors propose a model where a limited number of samples can be labelled and devise an algorithm to perform a powerful two-sample test in this setting.They then show desirable statistical properties of the test.

**Q2 Assessment Of The Paper:**

More detailed information regarding each of these aspects is given below:

**Q2(4) Quality Of Experiments (Optional):**

3: Good: The experimental evaluation is adequate, and the results convincingly support the main claims.

**Q2(5) Reproducibility:**

3: Good: Key resources (e.g., proofs, code, data) are available and key details (e.g., proofs, experimental setup) are sufficiently well-described for competent researchers to confidently reproduce the main results.

**Q3 Main Strengths:**

The main idea behind the paper is an original re-interpretation of a widely-used statistical technique. This or other similar tests are potentially widely-applicable to a variety of problems in different fields. The main ideas of the paper are explained clearly.

**Q4 Main Weakness:**

The theoretical results rely very strongly on the limiting behaviour presented in Theorem 2, where the optimal sampling distribution converges to a distribution defined on just two points. This seems to assume that one can resample an arbitrary number of times from these two features, when in reality this will hardly ever be possible. A more realistic setting would have been one where the class labels can only be sampled once per data point.  The authors also rely on Normal approximations, but no guideline is provided on what amount of data is sufficient.

**Q5 Detailed Comments To The Authors:**

The authors really should comment on what happens when we cannot resample a data point. In real-world settings, sampling from two small subregions of a distribution seems like a risky strategy, particularly if one is intere

- I found the optimization part of the proof of Theorem 2 (p.13) quite confusing. Is duality used there? The proof should be made more explicit.

-The notation in the proof of Thm 4 is confusing - isn't S_q the same as S_h?

- The discussion of the permutation test is vague and therefore hard to follow - it's hard to understand the precise nature of the test.

**Q7 Justification For Your Score:**

This is an interesting and important problem and the results are interesting and likely to spur more interest in this area. While I have some doubts about the specific setting, I think this is a valuable paper.

**Q9 Complying With Reviewing Instructions:**

1: Yes.

---

### Official Review · Reviewer_QEeY · 2022-04-13

**Q2(1) Originality/Novelty:** 3
**Q2(2) Significance/Impact:** 2
**Q2(3) Correctness/Technical Quality:** 3
**Q2(6) Clarity Of Writing:** 3
**Q6 Overall Score:** 6
**Q8 Confidence In Your Score:** 2

**Q1 Summary And Contributions:**

a two-sample test with limited sample label queries. In the first stage, the label is sampled uniformly to train a classifier to model the posterior distribution of labels. In the second stage, a bimodal query scheme is used to query labels from both classes. In the third stage, the Friedman-Rafsky two-sample test is conducted. Theoretical analysis and experimental results on several synthetical and real-world datasets have been shown.


**Q2 Assessment Of The Paper:**

More detailed information regarding each of these aspects is given below:

**Q2(4) Quality Of Experiments (Optional):**

2: Fair: The experimental evaluation is weak: important baselines are missing, or the results do not adequately support the main claims.

**Q2(5) Reproducibility:**

3: Good: Key resources (e.g., proofs, code, data) are available and key details (e.g., proofs, experimental setup) are sufficiently well-described for competent researchers to confidently reproduce the main results.

**Q3 Main Strengths:**

The paper tackles a novel setting, where a two-sample test needs to be conducted with limited labels. The paper is clearly written.  Theoretical results are given. Experiments and ablation studies are conducted.


**Q4 Main Weakness:**

The main concern I have about the paper is how to determine the sample n_t needed for the first stage. As the paper stated, to make the bimodal query scheme work, a good approximation of p(z|s) is needed, which depends on how many samples are used in the first stage. Is there a theoretical soundness way to determine how to choose n_t? Or like an experimental investigation of how to choose n_t in practice?

**Q5 Detailed Comments To The Authors:**

Curious to see could this method scale up to datasets with higher input dimensions or larger sample sizes? What would the bottleneck be in those situations?

**Q7 Justification For Your Score:**

While there are some minor questions, to my understanding, this is a novel method in a new setting. Also, the framework has been shown could extend to mo tests empirically.

**Q9 Complying With Reviewing Instructions:**

1: Yes.

---

### Decision · Program_Chairs · 2022-05-15

**Decision:**

Accept (Poster)

**Comment:**

Meta Review: This paper presents an approach to perform two-sample tests in an active learning-like setting where acquiring labels is expensive.

All of the reviewers agree that the quality of the paper is high. It addresses an interesting and practically-relevant settings; the proposed theoretical approach is novel, and it leads to a reasonable empirical algorithm. The paper is also well-written.

Pros
* Novel problem setting
* Clearly written
* Strong theoretical and experimental results

Cons
* Some choices, such as the number of initial required samples, are not clear in the paper.
* Some of the theoretical assumptions may not be entirely realistic, such as whether resampling is feasible and whether asymptotic guarantees are truly important for a limited sample size setting.